# A Schwarz iterative method to evaluate ocean-atmosphere coupling schemes; implementation and diagnostics in IPSL-CM6-SW-VLR

Olivier Marti[1], Sébastien Nguyen[1], Pascale Braconnot[1], Sophie Valcke[2], Florian Lemarié[3], and Eric Blayo[3]

[1]Laboratoire des Sciences du Climat et de l'Environnement, LSCE/IPSL, CEA-CNRS-UVSQ, Université Paris-Saclay, Gif-sur-Yvette, France
[2]CECI, Université de Toulouse, CNRS, CERFACS, Toulouse, France
[3]Univ. Grenoble Alpes, Inria, CNRS, Grenoble INP, LJK, 38000, Grenoble, France

**Correspondence:** Olivier Marti (olivier.marti@lsce.ipsl.fr)

**Abstract.** State-of-the-art Earth System models, like the ones used in the 6th Coupled Model Intercomparison Project (CMIP6), suffer from temporal inconsistencies at the ocean-atmosphere interface. Indeed, the coupling algorithms generally implemented in those models do not allow for a correct phasing between the ocean and the atmosphere, and hence between their diurnal cycle. A possibility to remove these temporal inconsistencies is to use an iterative coupling algorithm based on the Schwarz iterative method. Despite its large computational cost compared to standard coupling methods, which makes the algorithm implementation impractical as is for production runs, the Schwarz method is useful to evaluate some of the errors made in state-of-the-art ocean-atmosphere coupled models (e.g. in the representation of the processes related to diurnal cycle), as illustrated by the present study. IPSL-CM6-SW-VLR is a version of the low resolution version of IPSL-CM6 coupled model with a simplified land surface model, implementing a Schwarz iterative coupling scheme. Comparisons between coupled solutions obtained with this new scheme and the standard IPSL coupling scheme (referred to as *parallel* algorithm) show large differences after sunrise and before sunset, when the external forcing (insolation at top of atmosphere) has the fastest pace of change. At these times of the day, the difference between the two numerical solutions is often larger than $100\%$ of the solution, even with a small coupling period, thus suggesting that significant errors are potentially made with current coupling methods. Most of those differences can be strongly reduced by making only two iterations of the Schwarz method which leads to a doubling of the computing cost. Besides the *parallel* algorithm used in IPSL-CM6, we also test a so-called *sequential atmosphere-first* algorithm which is used in some coupled ocean-atmosphere models. We show that the *sequential* algorithm improves the numerical results compared to the *parallel* one, at the expanse of a loss of parallelism. The present study focuses on the ocean-atmosphere interface, with no sea ice. The problem with three components (ocean / sea ice / atmosphere) remains to be investigated.

## 1 Introduction

For historical and physical reasons, present-day coupling algorithms implemented in coupled general circulation models (CGCMs) are primarily driven by the necessity to conserve energy and mass at the air-sea interface. However the discretiza-

tion of the coupling problem often leads to inconsistencies in time and space associated to the coupling algorithm and to the grid-to-grid interpolation of air-sea fluxes and surface properties. In time, the coupling algorithms currently used in state of the art CGCMs do not provide the exact solution to the ocean-atmosphere problem, but an approximate one. Indeed, these approaches are mathematically inconsistent in the sense that they do not allow for a correct phasing between the ocean and the atmosphere. Roughly speaking, the existing coupling algorithms used in CGCMs split the total simulation time into smaller time intervals (called coupling periods) over which averaged-in-time boundary data are exchanged. The atmosphere computes the fluxes at the interface (heat, water and momentum) and the ocean computes the oceanic surface properties (water and sea ice temperatures, sea ice fraction, albedos, surface current). Two main algorithms are used, the *parallel* and the *sequential atmosphere-first* algorithm. In both methods, the interface fluxes for a coupling period are computed in the atmospheric model using the oceanic surface properties computed by the oceanic model and averaged over the previous coupling period. The two algorithms are *lagged*: there is a time lag (of one coupling period) between the model and its boundary conditions. They differ by the way atmospheric fluxes are used by the ocean. In the *parallel* algorithm, ocean and atmosphere run concurrently, which adds a level of parallelism and reduces the time to solution. During a coupling period, the ocean run uses the interface fluxes of the previous one, and compute the oceanic properties. Therefore, for a given coupling period, the fluxes used by the oceanic model are not coherent with the oceanic surface properties considered by the atmospheric model. In the *sequential atmosphere-first* algorithm, the atmosphere runs the coupling period while the ocean waits. This allows the ocean to use the fluxes of the present coupling period. The inconsistency is reduced, but not removed. The models can not run concurrently, which suppresses a level of parallelism, except in the case of a two-coupling-period lag (see the RPN model described below). The *parallel* algorithm has been implemented in many European CGCMs used in CMIP6 besides IPSL-CM6, for example in CNRM-CM6-1 developed by CNRM-CERFACS (Centre National de Recherches Météorologiques — Centre Européen de Recherche et de Formation Avancée en Calcul Scientifique), EC-Earth3 developed by a Europe-wide consortium of 27 research institutes from 10 European countries, MPI-ESM the Earth System Model developed by the Max-Planck-Institut für Meteorologie, or HadGEM3-GC31 set up by the UK MetOffice. The ocean-atmosphere coupling algorithm implemented in the CGCM developed at the European Centre for Medium-Range Weather Forecast (ECMWF) is quite different and involves three components, an atmosphere model, a wave model and an ocean model run sequentially in that order and therefore corresponds to the *sequential atmosphere-first* algorithm. The CGCM developed by RPN (Centre de Recherche en Prévision Numérique) from the Canadian meteorological and climatic services (Environment and Climate Change, Canada) also implements a *sequential atmosphere-first* algorithm, but with the particularity that the atmosphere receives and uses for one coupling period the surface properties calculated two coupling periods before by the ocean. This last algorithm allows to run the models concurrently and therefore to keep this level of parallelism, but increases the time lag and thus the inconsistency. To our knowledge, no model uses an *sequential ocean-first* algorithm.

Due to the overwhelming complexity of CGCMs, the consequences of inaccuracies in coupling algorithms on numerical solutions are hard to untangle, unless a properly (tightly) coupled solution can be used as a reference. Schwarz algorithms are attractive iterative coupling methods to cure the aforementioned temporal inconsistencies and provide tightly coupled solutions. As discussed in Lemarié (2008), the standard coupling methods correspond to one single iteration of a global-in-time iterative

Schwarz method. However, the theoretical analysis of the convergence properties of the Schwarz methods is restricted to relatively simple linear model problems (e.g. Gander et al., 1999; Gander and Halpern, 2007; Lemarié et al., 2013). More recently, Thery et al. (2020) analyzed the convergence for a coupled one-dimensional Ekman layer problem, with vertical profiles of viscosities in both fluids. But there is no a priory guarantee that the iterative process converges in practice when implemented in tri-dimensional ocean-atmosphere coupled models.

Preliminary numerical simulations using the Schwarz coupling method for the simulation of a tropical cyclone with a realistic regional coupled model have already been carried out by Lemarié et al. (2014). Ensemble simulations were designed by perturbations of the initial conditions and of the length of the coupling period. One ensemble was integrated using the Schwarz method and another using a *parallel* algorithm, as described previously. The Schwarz iterative coupling method led to a significantly reduced spread in the ensemble results (in terms of cyclone trajectory and intensity), thus suggesting that a source of error is removed with respect to the *parallel* coupling case. For these experiments the iterative process converges when coupling fully realistic numerical codes (Lemarié et al., 2014), which strengthens our belief that Schwarz methods can be a useful tool in the geophysical applications. Interestingly enough, a similar link between model uncertainties and consistency of the coupling method has been observed by Connors and Ganis (2011) on a coupled problem between two Navier-Stokes equations with interface conditions given by a bulk formulation.

The present study aims to assess the error made when using lagged coupling algorithms (*parallel* and *sequential*) in state-of-the art CGCMs. To do so, a mathematically consistent Schwarz iterative method is implemented in the IPSL Earth system model. It is used as a reference to evaluate the error due to the *lagged* algorithms. We study the convergence speed, compare the methods, and propose further developments in order to improve future ocean-atmosphere coupled models.

The paper is organized as follows. In section 2, we detail the *lagged* coupling algorithms, taking as an example the IPSL model, and the Schwarz iterative method. Section 3 describes the model and the experimental set-up. Section 4 analyses the results, in term of convergence speed and error assessment. Conclusion and future approaches are given in section 5.

## 2 State of the art of ocean-atmosphere coupling algorithms and the Schwarz method

Multiphysics coupling methods used in the context of Earth System models can be classified into two general categories (e.g. Lemarié et al., 2015; Gross et al., 2018). The first one (usually referred to as asynchronous coupling, and called *lagged* in the present paper[1]) is based on an exchange of average fluxes between the models. The second one (referred to as synchronous coupling in Lemarié et al. (2015)) uses instantaneous fluxes. Climate modelling focuses primarily on how energy is exchanged between the Earth and the outer space, and is transported by the ocean and the atmosphere. When designing a coupling method in the context of CGCMs, water and energy conservation at the machine precision are the key features. Those conservation principles are impossible to satisfy when exchanging instantaneous fluxes. Coupled ocean-atmosphere models used for long-term integration (decades to millennia) all use a coupling methodology based on the exchange of time averaged or time integrated fluxes.

---

[1]The terms "synchronous" and "asynchronous" may have a totally different signification for climate modellers, and we prefer to avoid them.

## 2.1 Current ocean-atmosphere coupling in IPSL-CM6: the legacy *parallel* algorithm

The top panel (a) of Fig. 1 describes how quantities are exchanged between the ocean and the atmosphere in the IPSL climate model from 1997 to now (Braconnot et al., 1997; Marti et al., 2010; Dufresne et al., 2013; Sepulchre et al., 2020; Boucher et al., 2020) knowing that both models are run in a *parallel* way. The coupling period $\Delta t$ (which should not be confused with the dynamical time-step in the individual models) typically varies between $1\,\mathrm{hour}$ to $1\,\mathrm{day}$, depending on the configuration and the model generation. Ocean and atmosphere dynamical time steps are always smaller, but commensurable with the coupling period. To describe this coupling strategy, we introduce the atmospheric state vector $\mathcal{A}$ (encompassing temperature, humidity, pressure, velocity...) and the oceanic state vector $\mathcal{O}$ (encompassing temperature, salinity, velocity ...). The time evolution of the atmosphere and the ocean is symbolically described by

$$\frac{d\mathcal{A}}{dt} = \mathbf{F}_{\mathcal{A}}(\mathcal{A}, \mathbf{f}_{\Omega}), \qquad \frac{d\mathcal{O}}{dt} = \mathbf{F}_{\mathcal{O}}(\mathcal{O}, \mathbf{f}_{\Omega}) \tag{1}$$

where $\mathbf{F}_{\mathcal{A}}$ and $\mathbf{F}_{\mathcal{O}}$ are partial differential operators including parameterizations, and $\mathbf{f}_{\Omega}$ represents the fluxes at the ocean-atmosphere interface $\Omega$. This formulation is symmetric between the ocean and the atmosphere. But, in practice, in CGCMs the symmetry is broken between the fast atmospheric component and the slower oceanic component. The fluxes are generally computed by the atmospheric component or by an interface model, using oceanic surface quantities and atmospheric quantities taken in the vicinity of the air-sea interface (sea-surface properties are noted $\mathcal{O}_{\Omega}$ in the following), meaning that (1) can be reformulated as

$$\frac{d\mathcal{A}}{dt} = \mathbf{F}_{\mathcal{A}}(\mathcal{A}, \mathcal{O}_{\Omega}), \qquad \frac{d\mathcal{O}}{dt} = \mathbf{F}_{\mathcal{O}}(\mathcal{O}, \mathbf{f}_{\Omega}), \qquad \mathbf{f}_{\Omega} = \mathbf{f}_{\Omega}(\mathcal{A}, \mathcal{O}_{\Omega}) \tag{2}$$

With such an approach, the atmospheric model receives surface properties like sea surface and sea ice surface temperature, fraction of sea ice, albedo and velocities of the surfaces (sea water and sea ice) and computes its own interfacial fluxes which are then sent to the oceanic component. Interfacial fluxes sent by the atmosphere include heat fluxes (radiative and turbulent), water fluxes (solid and liquid precipitation, evaporation, sublimation) and momentum fluxes (wind stress).

As mentioned earlier, the coupling algorithm in the IPSL climate model is based on an exchange of averaged-in-time fluxes. We define $\langle\ldots\rangle_{t_1}^{t_2}$ as the time average in the interval $[t_1, t_2]$, and $\Delta t$ the coupling period. A schematic view of the exchanges between the ocean and the atmosphere is given in Fig. 1. To run over a coupling period $\Delta t$, each component uses the available boundary conditions which are time averaged from the previous coupling period. We thus have:

$$\left.\frac{d\mathcal{A}}{dt}\right|_{t}^{t+\Delta t} = \mathbf{F}_{\mathcal{A}}(\mathcal{A}, \langle\mathcal{O}_{\Omega}\rangle_{t-\Delta t}^{t}), \qquad \left.\frac{d\mathcal{O}}{dt}\right|_{t}^{t+\Delta t} = \mathbf{F}_{\mathcal{O}}(\mathcal{O}, \langle\mathbf{f}_{\Omega}\rangle_{t-\Delta t}^{t}) \tag{3}$$

To be more precise, the fluxes sent from the atmosphere to the ocean and the surface properties sent from the ocean to the atmosphere at time $t$ are :

$$\langle\mathbf{f}_{\Omega}\rangle_{t-\Delta t}^{t} = \left\langle \mathbf{f}_{\Omega}(\mathcal{A}, \langle\mathcal{O}_{\Omega}\rangle_{t-2\Delta t}^{t-\Delta t}) \right\rangle_{t-\Delta t}^{t}, \qquad \langle\mathcal{O}_{\Omega}\rangle_{t-\Delta t}^{t} = \left\langle \mathcal{O}_{\Omega}(\mathcal{O}, \langle\mathbf{f}_{\Omega}\rangle_{t-2\Delta t}^{t-\Delta t}) \right\rangle_{t-\Delta t}^{t} \tag{4}$$

**Figure 1.** Time stencil of the exchanges between the ocean and the atmosphere in the lagged algorithms. a) The *parallel* algorithm, redrawn from Fig. 5.4 of Lemarié (2008). b) The *sequential atmosphere-first* algorithm.

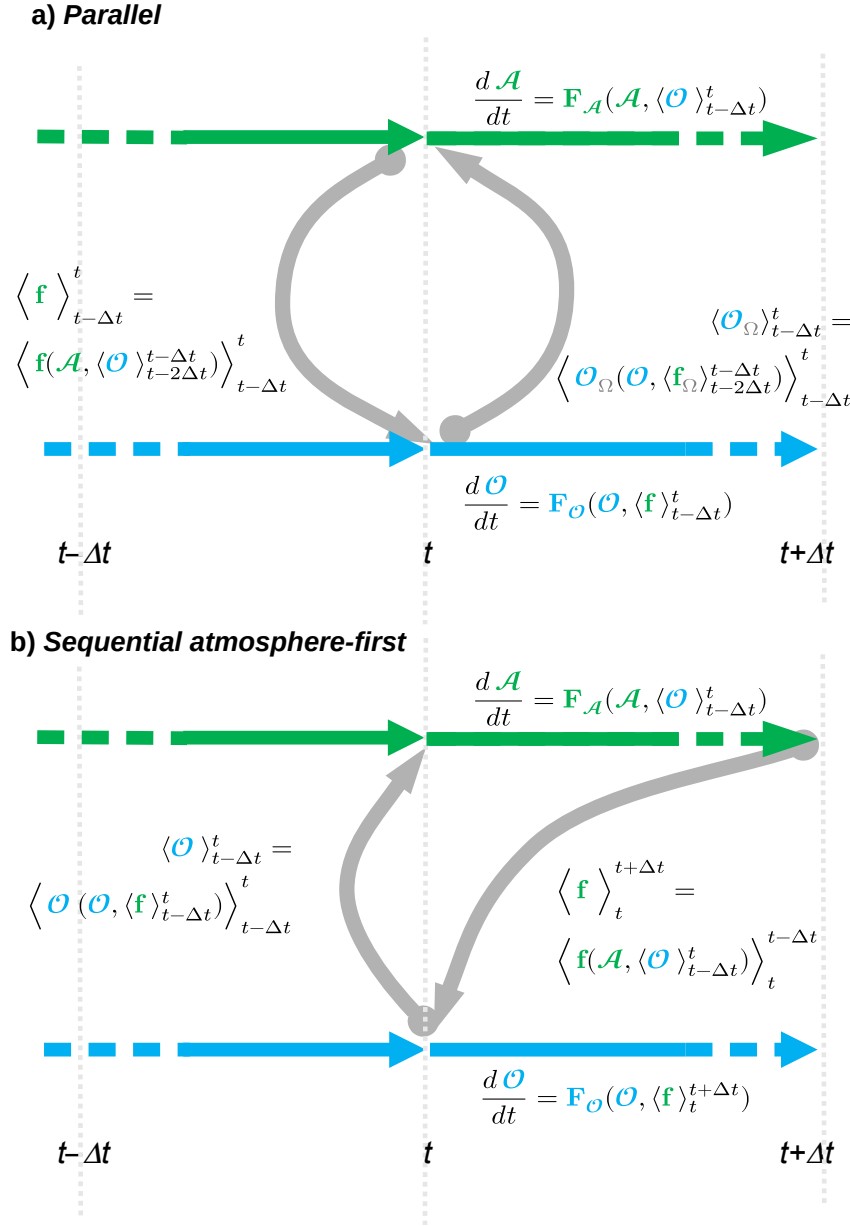

Substituting (4) in (3), we thus can write the evolution of the ocean $\mathcal{O}$ and the atmosphere $\mathcal{A}$ from $t$ to $t + \Delta t$ as :

$$120 \quad \frac{d\mathcal{O}}{dt}\bigg|_t^{t+\Delta t} = \mathbf{F}_{\mathcal{O}}\left(\mathcal{O}, \left\langle \mathbf{f}(\mathcal{A}, \langle \mathcal{O}_\Omega \rangle_{t-2\Delta t}^{t-\Delta t})\right\rangle_{t-\Delta t}^{t}\right), \qquad \frac{d\mathcal{A}}{dt}\bigg|_t^{t+\Delta t} = \mathbf{F}_{\mathcal{A}}\left(\mathcal{A}, \left\langle \mathcal{O}_\Omega(\mathcal{O}, \langle \mathbf{f}_\Omega \rangle_{t-2\Delta t}^{t-\Delta t})\right\rangle_{t-\Delta t}^{t+\Delta t}\right) \qquad (5)$$

The interfacial flux used as a boundary condition for the ocean between $[t, t + \Delta t]$ is computed by the atmosphere using sea-surface values of the ocean from the time range $[t - 2\Delta t, t - \Delta t]$. Symmetrically, the sea-surface properties used to run the atmosphere during the time range $[t, t + \Delta t]$ are computed using surface values of the ocean from the time range $[t - 2\Delta t, t - \Delta t]$. Eq. (4) and Eq. (5) demonstrate the time shift between the two models, and how the boundary conditions lag the models. The numerical solution thus obtained is not mathematically consistent and suffers from synchronicity issues which ultimately may yield the numerical implementation to be unstable in the sense that the error compared to the exact solution keeps increasing with time.

## 2.2 The *sequential atmosphere-first* algorithm

The bottom panel (b) of Fig. 1 describes how quantities are exchanged between the ocean and the atmosphere in the *atmosphere-first* algorithm. The evolution of ocean $\mathcal{O}$ and atmosphere $\mathcal{A}$ become :

$$\left.\frac{d\mathcal{A}}{dt}\right|_t^{t+\Delta t} = \mathbf{F}_{\mathcal{A}}(\mathcal{A}, \langle \mathcal{O}_\Omega \rangle_{t-\Delta t}^t), \qquad \left.\frac{d\mathcal{O}}{dt}\right|_t^{t+\Delta t} = \mathbf{F}_{\mathcal{O}}(\mathcal{O}, \langle \mathbf{f}_\Omega \rangle_t^{t+\Delta t}) \tag{6}$$

and

$$\langle \mathbf{f}_\Omega \rangle_{t-\Delta t}^t = \left\langle \mathbf{f}_\Omega(\mathcal{A}, \langle \mathcal{O}_\Omega \rangle_{t-2\Delta t}^{t-\Delta t}) \right\rangle_{t-\Delta t}^t, \qquad \langle \mathcal{O}_\Omega \rangle_{t-\Delta t}^t = \left\langle \mathcal{O}_\Omega(\mathcal{O}, \langle \mathbf{f}_\Omega \rangle_{t-\Delta t}^t) \right\rangle_{t-\Delta t}^t \tag{7}$$

Substituting (7) in (6), we now have an asymmetry of the evolution of the ocean $\mathcal{O}$ and the atmosphere $\mathcal{A}$ from $t$ to $t + \Delta t$ :

$$\left.\frac{d\mathcal{O}}{dt}\right|_t^{t+\Delta t} = \mathbf{F}_{\mathcal{O}}\left( \mathcal{O}, \left\langle \mathbf{f}(\mathcal{A}, \langle \mathcal{O}_\Omega \rangle_{t-\Delta t}^t) \right\rangle_t^{t+\Delta t} \right), \qquad \left.\frac{d\mathcal{A}}{dt}\right|_t^{t+\Delta t} = \mathbf{F}_{\mathcal{A}}\left( \mathcal{A}, \left\langle \mathcal{O}_\Omega(\mathcal{O}, \langle \mathbf{f}_\Omega \rangle_{t-\Delta t}^t) \right\rangle_{t-\Delta t}^t \right) \tag{8}$$

This *atmosphere-first* algorithm has been easily implemented in IPSL-CM6 by changing lag parameters in the OASIS3-MCT coupler namelist. The symmetric *ocean-first* has been also implemented, but is not detailed here.

To our knowledge, no coupled ocean-atmosphere model uses a coupling algorithm that is fully mathematically consistent. The survey of actual use cases (Valcke, personal communication) presented in the introduction shows that they all induce a time lag between the models and the boundary conditions, either in both directions (double sided lag) or at least in one direction (single sided lag). In the GFDL Earth system model, the FMS coupler offers the possibility to use an implicit scheme to compute the interface quantities. But only the vertical turbulent diffusion part of the ocean and atmosphere models are considered (Balaji et al., 2006), and the full model equations are not synchronised.

## 2.3 The Schwarz iterative method

The Schwarz iterative method is described and analyzed in Lemarié et al. (2015) in the context of ocean-atmosphere coupling. The basic idea is to separate the global coupled problem on $\mathcal{A} \cup \mathcal{O}$ into separated sub-problems on $\mathcal{A}$ and $\mathcal{O}$, which can be solved separately with an appropriate exchange of boundary conditions at the common interface $\Omega$. An iterative process

**Figure 2.** Stencil of the Schwarz iterative method, shown for the *parallel* algorithm. $k$ is the iteration index. The $^\star$ superscript denotes the converged solution. At each iteration $k$, the first guesses of $\mathcal{A}$ and $\mathcal{O}$ at time $t$ are taken from the states of $\mathcal{A}$ and $\mathcal{O}$ at the end of the previous Schwarz window $[t - \Delta t, t]$, and for the last iteration. Only the boundary conditions are updated at each iteration.

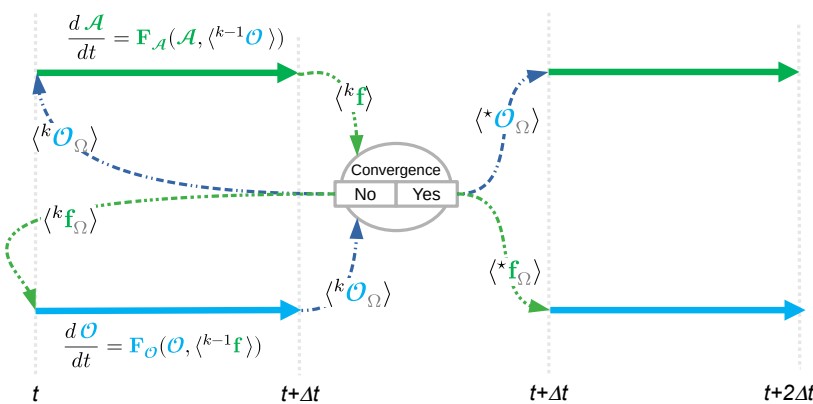

is applied to achieve the convergence to the solution of the global problem. The main concern about this approach is the computational cost which directly depends on the convergence speed. As illustrated in Fig. 2, we iterate the system until
convergence over the time interval $[t, t + \Delta t]$. The first guesses of $\mathcal{A}$ and $\mathcal{O}$ at time $t$ are taken from the states of $\mathcal{A}$ and $\mathcal{O}$ at the end of the previous coupling period $[t - \Delta t, t]$. The iterative process from iteration $k - 1$ to iteration $k$ is described by:

$$\begin{cases} \dfrac{d\,^k\mathcal{A}}{dt} = \mathbf{F}_{\mathcal{A}}(\,^k\mathcal{A}, \langle\,^{k-1}\mathcal{O}_\Omega\rangle) \\[2mm] \dfrac{d\,^k\mathcal{O}}{dt} = \mathbf{F}_{\mathcal{O}}(\,^k\mathcal{O}, \langle\,^{k-1}\mathbf{f}_\Omega\rangle) \\[2mm] \,^k\mathbf{f}_\Omega = \mathbf{f}_\Omega(\,^{k-1}\mathcal{A}, \langle\,^{k-2}\mathcal{O}_\Omega\rangle) \\[2mm] \,^k\mathcal{O}_\Omega = \mathcal{O}_\Omega(\,^{k-1}\mathcal{O}, \langle\,^{k-2}\mathbf{f}_\Omega\rangle) \end{cases} \tag{9}$$

In the classification of domain decomposition methods, such a Schwarz algorithm applied to the *parallel* coupling algorithm is called a *parallel* (or *additive*) Schwarz method, since it allows the concurrent resolution of the first two equations of (9).
The Schwarz method applied to a *sequential* coupling simply consists in replacing the index $k - 1$ by the index $k$ in one of the first two equations. One then obtains a so-called *sequential* (or *multiplicative*) Schwarz method, which imposes that the equation using the information at iteration $k - 1$ be resolved first, allowing then the resolution of the equation in the other medium. This *sequential* algorithm requires thus about twice the elapsed time of the concurrent version (if one considers that

the elapsed times for each medium are balanced and that the two medias run on different sets of processors or cores). However it is well-known (and easy to prove) that, in linear cases, the *sequential* algorithm requires generally approximately twice less iterations to converge than the *parallel* algorithm.

For a state-of-the-art CGCMs with complex parameterizations, we have no mathematical evidence that the algorithm converges. Indeed, as mentioned in Keyes et al. (2013), reaching a tight coupling between the components to be coupled requires smoothness. However, both ocean and atmosphere models include parameterizations that are are potentially not differentiable. This is for instance the case of the bulk formulas used to compute the turbulent fluxes at the air-sea interface (e.g. Pelletier et al., 2018). A first step is thus to test the convergence when coupling realistic models. Assuming that the algorithm converges, for large values of $k$ we would have $^{k-1}(\mathcal{A} \cup \mathcal{O}) = {}^{k}(\mathcal{A} \cup \mathcal{O}) = {}^{\star}(\mathcal{A} \cup \mathcal{O})$, with the left superscript $^\star$ denoting the converged solution. The evolution of $^{\star}\mathcal{O}$ and $^{\star}\mathcal{A}$ is given by:

$$\frac{d\,{}^{\star}\mathcal{O}}{dt}\bigg|_t^{t+\Delta t} = \mathbf{F}_{\mathcal{O}}\left({}^{\star}\mathcal{O}, \left\langle \mathbf{f}({}^{\star}\mathcal{A}, \langle {}^{\star}\mathcal{O}_{\Omega}\rangle_t^{t+\Delta t})\right\rangle_t^{t+\Delta t}\right), \qquad \frac{d\,{}^{\star}\mathcal{A}}{dt}\bigg|_t^{t+\Delta t} = \mathbf{F}_{\mathcal{A}}\left({}^{\star}\mathcal{A}, \left\langle {}^{\star}\mathcal{O}_{\Omega}({}^{\star}\mathcal{O}, \langle {}^{\star}\mathbf{f}_{\Omega}\rangle_t^{t+\Delta t})\right\rangle_t^{t+\Delta t}\right) \quad (10)$$

where it is clear that models and boundary conditions are now fully synchronized, meaning that the algorithm is mathematically consistent. In simple linear models, the unicity of the converged solution is proven. It does not depend on the initial guess which can be chosen arbitrarily. It also does not depend on the coupling algorithm: *parallel* and the *sequential* algorithms yield the same solution. However, different initial states will change the convergence speed. In our case, the models are strongly non linear. The coupled problem may have several solutions, an the converged solution may depend on the initial guess. A relevant choice of the initial guess is then important. We use what is the most simple and obvious choice: the converged solution of the previous Schwarz window.

The Schwarz iterative procedure may span several coupling periods. The time interval $[t, t + \Delta t]$ is then called the 'Schwarz window'. It is divided in $p$ coupling periods. At the end of each Schwarz window, the models send the boundary conditions as a vector of values for the coupling intervals $[t, t + \frac{1}{p}\Delta t]$, $[t + \frac{1}{p}\Delta t, t + \frac{2}{p}\Delta t]$, ..., $[t + \frac{p-1}{p}\Delta t, t + \Delta t]$. The boundaries conditions exchanged between the models are then vector of quantities :

$$\mathbf{f}_{\Omega} = \{\langle \mathbf{f}_{\Omega}\rangle_t^{t+\frac{1}{p}\Delta t}, \langle \mathbf{f}_{\Omega}\rangle_{t+\frac{1}{p}\Delta t}^{t+\frac{2}{p}\Delta t}, \dots, \langle \mathbf{f}_{\Omega}\rangle_{t+\frac{p-1}{p}\Delta t}^{t+\Delta t}\}, \qquad \mathcal{O}_{\Omega} = \{\langle \mathcal{O}_{\Omega}\rangle_t^{t+\frac{1}{p}\Delta t}, \langle \mathcal{O}_{\Omega}\rangle_{t+\frac{1}{p}\Delta t}^{t+\frac{2}{p}\Delta t}, \dots, \langle \mathcal{O}_{\Omega}\rangle_{t+\frac{p-1}{p}\Delta t}^{t+\Delta t}\} \quad (11)$$

With this method, the frequency of exchange can be different for each field, provided that the coupling period of each field is a whole division of the Schwarz window (value of $p$ is specific to each field). With more than two models, the Schwarz method can be used to couple models by pairs, for the whole system, or for any relevant decomposition of the system. More details about the technical implementation in an Earth System Model are given in section 3.2. The following study handles only the case where the Schwarz window is equal to the coupling period ($p = 1$). The possibility to have a longer Schwarz window has not been coded for the sake of simplicity. Also, we did not test the algorithm with a lag of two coupling periods, as it would have been quite difficult to implement technically.

**Table 1.** IPSL-CM6-SW-VLR compared to IPSLCM5-A2-LR and IPSL-CM6

| Characteristics | Comment and reference |
| --- | --- |
| Code version | Same as IPSL-CM6 (Boucher et al., 2020). |
| Résolution | Same as IPSLCM5-A2-LR (Dufresne et al., 2013; Sepulchre et al., 2020) for ocean and atmosphere. |
| Atmospheric and ocean physics | Same as IPSLCM5-A2-LR (Marti et al., 2010; Dufresne et al., 2013; Sepulchre et al., 2020). |
| Parameter tuning (atmosphere) | Method described in Sepulchre et al. (2020) |
| Land surface scheme | Bucket (Ducoudré et al., 1993). IPSLCM5A2 IPSLCM6 uses ORCHIDEE (Ducoudré et al., 1993). |
| Sea ice scheme | LIM3 mono-category (Rousset et al., 2015). |

## 3    Model and experiments

### 3.1    The IPSL-CM6-SW-VLR version of the IPSL Earth system model

At the start of this study, IPSL had two operational Earth System Models available, IPSL-CM5A2-LR and IPSL-CM6-LR. IPSL-CM5A2-LR is an upgrade of IPSL-CM5A-LR (Marti et al., 2010; Dufresne et al., 2013) used by IPSL for the CMIP5 intercomparison exercise, set up by Sepulchre et al. (2020). Compared to IPSL-CM5A-LR, the atmospheric model is tuned to reduce the surface cold bias and enhance the Atlantic meridional overturning circulation. The atmospheric code includes a supplemental level of shared memory parallelization that strongly improves the model scalability and speed. This model has an atmospheric resolution of 3.75° x 1.875° in longitude-latitude and 39 vertical levels. It has an oceanic resolution of 2 degrees and 31 vertical levels in the ocean. It runs at 70 simulated years per wall-clock day.

IPSL-CM6-LR (Boucher et al., 2020) is the model used by IPSL for the CMIP6 intercomparison exercise. It has a higher resolution in both ocean and atmosphere. All components (ocean, atmosphere, sea ice and land surface) have been improved with better physics compared to IPSL-CM5A2-LR. It runs at 10 simulated years per wall-clock day. IPSL-CM6-LR computer code and running environment brings to the user a strong improvement in terms of performance, portability, readability, versatility and quality control. See Boucher et al. (2020) for details.

The present study uses the codes of IPSL-CM6, but runs at the resolution of IPSL-CM5A2-LR. As an iterative Schwarz method strongly increases the computing time, the choice of a low resolution allows to contain the computing cost. As we planned high difficulties to implement the Schwarz method in the old style coding of IPSL-CM5A2-LR, the choice of the newer code appeared obvious.

The parameters of the atmospheric model allow to reproduce exactly the atmosphere of IPSL-CM5A2-LR when atmosphere is run in standalone mode. In the ocean, the sea ice model LIM3 is used with one category of ice (IPSL-CM6-LR uses 5 ice categories, based on ice thickness; see Rousset et al. (2015) for more details for LIM3 in mono category). The land surface model ORCHIDEE was removed to simplify and speed up the implementation of the Schwarz algorithm. As a soil model, we use the simple bucket model included in the atmosphere code (Ducoudré et al., 1993). The specificity of IPSL-CM6-SW-VLR with respect to IPSLCM5-A2-LR and IPSLCM6 are given in Table 1.

This specific version of the model is called IPSL-CM6-SW-VLR for further reference, SW standing for Schwarz. A short evaluation of the performance of IPSL-CM6-SW-VLR is given in Appendix A.

### 3.2 Implementation of the Schwarz algorithm in IPSL-CM6

The base of the Schwarz iterative algorithm is to repeat each Schwarz window with the same initial condition for each iteration, but with changing boundary conditions at the ocean-atmosphere interface $\Omega$ (the ones produced by the previous iteration). IPSL-CMs are restartable models: they produce the same result (bitwise) when run in one chunk, or when the run is split in small chunks, with the final state of each chunk written to disk and read by the following one. In the ocean and atmosphere codes, we implement the possibility to save/restore the fields needed for a restart to/from the computer memory.

The time loop of the models are replaced by three nested loops. The outer one loops on coupling periods. The middle one loops over Schwarz iterations. The inner one loops over the model time steps inside a coupling period. (For a coupling period of $\Delta t = 1\,\mathrm{h}$, the ocean performs 2 time steps and the atmosphere 6 time steps for the vertical physics, 30 for the dynamics and 1 for the radiation scheme). At the first Schwarz iteration of a Schwarz window, the initial states of the atmosphere $\mathcal{A}$ and the ocean $\mathcal{O}$ is the final state from the previous Schwarz window, once the Schwarz iterations have converged. This state is saved in memory, and will be read at the beginning of each iteration to initialize $\mathcal{A}$ and $\mathcal{O}$ with the same state for each Schwarz iteration. At the end of each iteration, the boundary conditions are sent to the companion model for use during the next one. The boundary conditions evolve during the iterative process. In this implementation, the length of the Schwarz window must equal the coupling period. The details of the different loops are given in Appendix B.

### 3.3 Experiments

We have run three sets of experiments (see Table 2). The first set uses the *parallel* algorithm. The second set uses the *atmosphere-first* algorithm. A third set uses the *ocean-first* algorithm. This last method is of no interest for operational use of climate model, but helped us to analyse some of the results. For each set, we run two experiments, with coupling periods of $\Delta t = 1\,\mathrm{h}$ and $\Delta t = 4\,\mathrm{h}$. The number of iterations is fixed to 50. The coupling fields exchanged between the models are written out at all iterations by the coupler OASIS, which allow us to study the convergence. Experiments are 5-day long (*i.e.* 120 and 30 Schwarz windows, or coupling periods in this case). The initial state is the end of a fifty year control experiment with pre-industrial forcings, run with the non-iterated *parallel* algorithm.

## 4 Results

### 4.1 Convergence

Figure 3 shows the behaviour of the sea surface temperature $T_\Omega$ along the iterative process for four selected cases in time and space for the *parallel* algorithm. These cases represent typical behaviours. The yellow dots show the values at the end of the previous Schwarz window. This is the initial state of the Schwarz iterations for the present coupling period. The green dots

**Table 2.** Main characteristics of experiments

| Name | Coupling period | Coupling algorithm |
|------|-----------------|--------------------|
| Sw1h50i | $\Delta t = 1\,\mathrm{h}$ | *Parallel* |
| Sw4h50i | $\Delta t = 4\,\mathrm{h}$ | *Parallel* |
| Sw1h50iA | $\Delta t = 1\,\mathrm{h}$ | *atmosphere-first* |
| Sw4h50iA | $\Delta t = 4\,\mathrm{h}$ | *atmosphere-first* |
| Sw1h50iO | $\Delta t = 1\,\mathrm{h}$ | *ocean-first* |
| Sw4h50iO | $\Delta t = 4\,\mathrm{h}$ | *ocean-first* |

**Figure 3.** Behaviour of the sea surface temperature for four selected cases (i.e. instances of the Schwarz algorithm in space $\times$ time), for the *parallel* algorithm. For each graph, the yellow dots show the values at the end of the previous Schwarz window, when Schwarz has converged. This is the initial state of the present window. The green dots show the values after the first iteration. It is the value that the models use with the legacy *parallel* algorithm not iterated. The blue dots show the iterative process. Dots become grey when $T_\Omega$ is considered to be converged. The two top cases (a) and (b) come from the $\Delta t = 4\,\mathrm{h}$ experiment. The bottom cases (c) and (d) come from the $\Delta t = 1\,\mathrm{h}$ experiment.

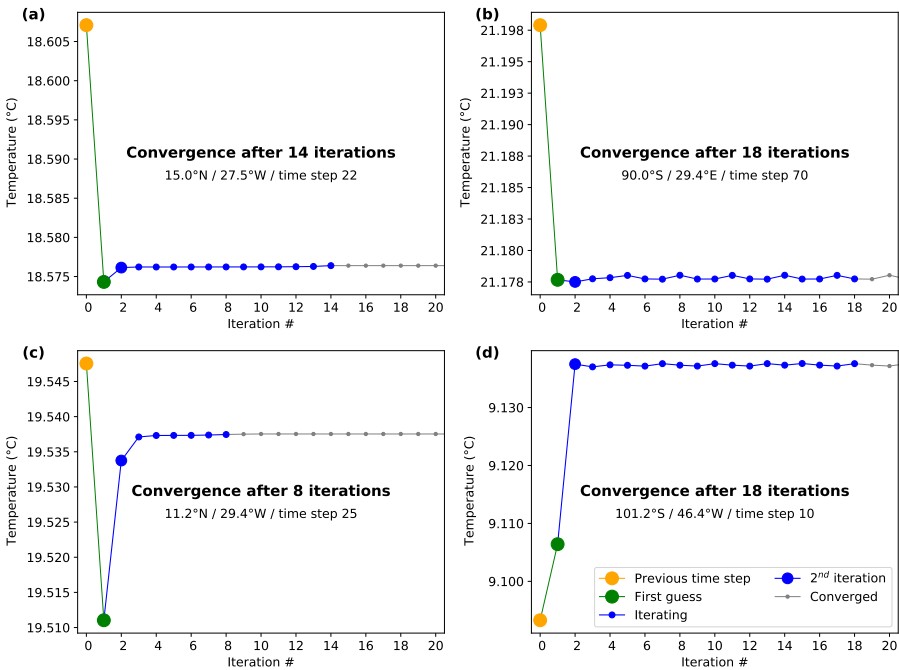

show the values after the first iteration. It is the values that the models would compute without Schwarz. The blue dots show the iterative process. Dots become grey when $T_\Omega$ is considered to be converged.

To decide if the convergence is reached at iteration $k_{conv}$, we consider $A_{T_\Omega}(k_{conv})$, the amplitude of the $T_\Omega$ changes after the iteration $k_{conv}$. $A_{T_\Omega}(k_{conv}) = \max_{k=k_{conv}}^{k=50}(T_\Omega) - \min_{k=k_{conv}}^{k=50}(T_\Omega)$. The convergence criterion is fulfilled if one of the following condition is met :

- Oscillation from iteration $k_{conv}$ to 50 has an amplitude which is negligible compared to the total range of the signal, i.e. if $A_{T_\Omega}(k_{conv}) \leq 10^{-3} A_{T_\Omega}$.

- Final oscillation from iteration $k = k_{conv}$ to $k = 50$ has an amplitude $A_\Omega(k_{conv})$ always lower than $10^{-4}\,°\mathrm{C}$ for temperature, $10^{-2}\,\mathrm{W m^{-2}}$ for heat fluxes.

- Oscillation has an amplitude from iteration $k_{conv}$ to 50 which is not bigger than the amplitude from iteration 41 to 50, i.e $A_{T_\Omega}(k_{conv}) \leq A_{T_\Omega}(40)$. This last criterion supposes that convergence is always reached at iteration $k = 40$. For points free of the sea ice, this criterion is not necessary, as one of the two above is always verified.

The speed of convergence is sensitive to the definition of these criteria, which mostly come from a *rule of the thumb* rather than from a rigorous mathematical analysis. A small residual oscillation is observed in all cases. The mathematics of the Schwarz method for the ocean-atmosphere coupling has been developed in Lemarié (2008), Lemarié et al. (2014, 2015) and Thery et al. (2020). The theory is robust and well established for two fluids with fixed turbulent viscosities. We have no theoretical frame when a third medium, sea ice in our case, is present. In all of the following, we will not analyse the
behaviour of the model when sea ice is present, and study only ice-free points. Text and figures present the behaviour of the sea-surface temperature. When the sea-surface temperature has converged, the atmosphere sees the same boundary condition at each iteration, and compute the same fluxes. The converged solution computed in Eq. (10) is theoretically the same for the three algorithms (*parallel* and *sequential*). This means that the results should be the same for all experiments with the same time step. But the convergence is not fully reached. A small oscillation remains. That means that at the end of the first Schwarz
window, the solution is specific for each experiment. As small as it is, this difference explains why the experiments follow different trajectories, the climate being chaotic.

     Figure 4 shows an histogram of the number of iterations for all experiments. We consider all the instances of the iterative procedures, for each ocean point of the atmosphere grid, and for all Schwarz windows. As explained above, we consider only points with no sea ice. In the *parallel*-$\Delta t = 1\,\mathrm{h}$ experiment, the Schwarz algorithm converges at the first iteration in almost
20 % of the cases. Two iterations are enough in almost 80 % of the cases. Only of few percents of cases require more iterations. The *ocean-first* slightly improve the result by a few percents. The *atmosphere-first* algorithm shows convergence at the first iteration for almost 100 % of the cases.

     For the $\Delta t = 4\,\mathrm{h}$ experiments, convergence is rarely reached in 1 iteration. In most of the cases 2 to 4 iterations are required. We still observed that the *parallel* and the *ocean-first* algorithms yield close results, the second one being faster. The *atmo-*
*sphere-first* strongly improves the speed of convergence.

     But the number of iterations might be sensitive to the choice of the convergence criterion. By construction, the convergence speed is in theory identical for all variables. After SST convergence, the atmosphere uses the same values of SST at each

**Figure 4.** Number of iterations for convergence for the (a, top) $\Delta t = 1\,\text{h}$ and (b, bottom) $\Delta t = 4\,\text{h}$ experiments. The total number of cases is $536,800 = 120$ Schwarz windows $\times 4,553$ ice free grid points in the $\Delta t = 1\,\text{h}$ experiments. And $136,800 = 30$ Schwarz windows $\times 4,560$ ice free grid points in the $\Delta t = 4\,\text{h}$ experiments. This number of cases is given for the *parallel* algorithm, and slightly differs for the other algorithms. The ordinates show the number of cases in percentage of the total number of cases in time $\times$ space.

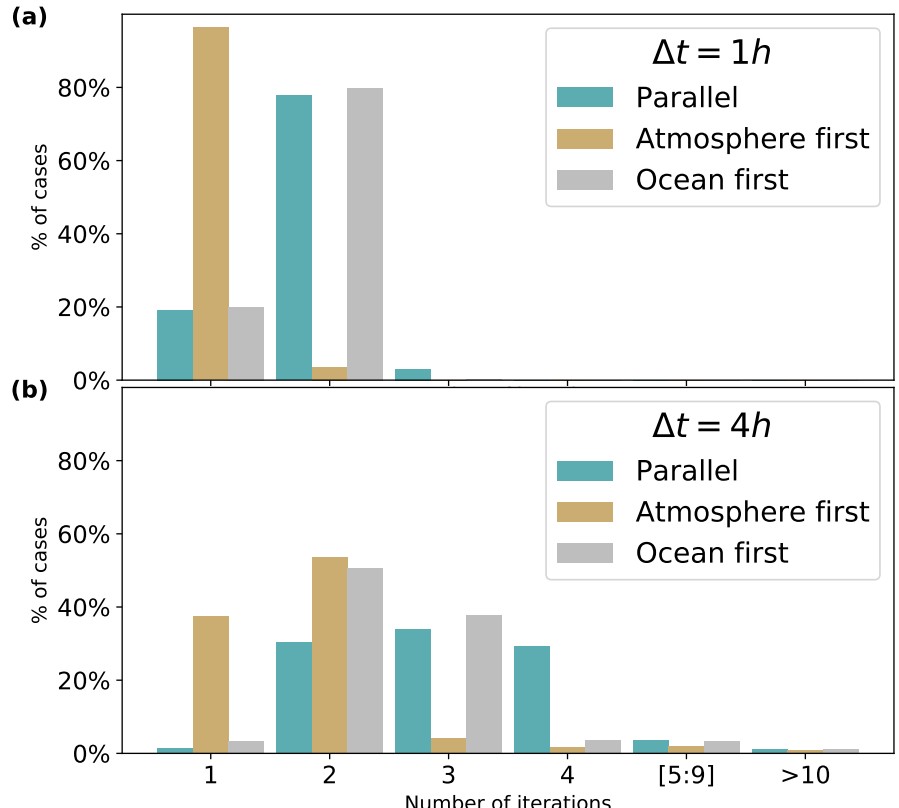

iteration, and computes the same fluxes. Symmetrically, when the fluxes computed by atmosphere have converged, the ocean can do nothing but producing the same SST at each iteration. In practice, the full convergence is not obtained, with a small oscillation of the values. As the convergence criterion is somewhat arbitrary, the computation of the number of iterations before convergence can give different values for the different variables. In the following, we diagnose the difference between the solutions with and without Schwarz, which does not depend on an arbitrary criterion.

## 4.2 Diagnosing the error of lagged coupling

Figure 5 shows the relative error in the change of sea surface temperature during one coupling period when the Schwarz method is not used. The error is computed on the sea surface temperature (SST) trend during a coupling period. At each Schwarz iteration, the model computes an occurrence of the SST trend. At the first iteration, the trend is the one that the model would calculate with the legacy lagged coupling. We can then compare it with the trend obtained after convergence. This

**Figure 5.** Relative error of the change of sea surface temperature during a Schwarz window. The error is computed as the ratio between i) the correction due to the iterative procedure (the jump from green dot to converged solution in grey in Fig. 3) and ii) the solution change between $t$ to $t + \Delta t$ with no Schwarz iteration (the jump from yellow to green dots in Fig. 3). See legend for Fig. 4 for the explanation of ordinate axis.

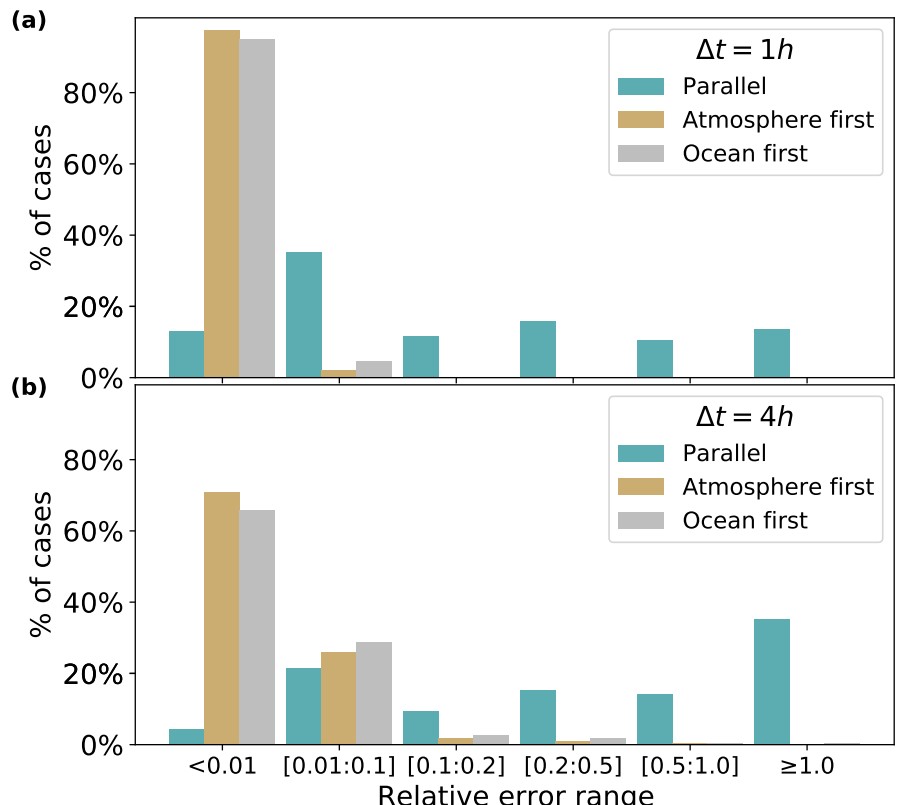

comparison of the two terms is done on a unique trajectory of the model. This trajectory uses the trend obtained at the last iterations. The error is computed as the ratio between i) the correction due to the iterative procedure (the jump from green dots to converged solution in grey in Fig. 3) and ii) the solution change between $t$ to $t + \Delta t$ with no Schwarz iteration (the jump from green to yellow dots in Fig. 3).

In the *parallel*-$\Delta t = 1\,\mathrm{h}$ experiment, the relative error is negligible (less than 0.01) in about 15 % of the cases. It is small (less than 0.1) in almost 50 % of the cases. But it is larger than 0.1 for the other half. The relative error is even larger than 0.5 in 25 % of the cases. The *atmosphere-first* shows strongly improved results, with a negligible error for 97 % of the points. The conclusion for experiment *ocean-first* is somewhat different from what the histogram of iterations (Fig. 4) shows. The results are very close to the *atmosphere-first* experiments. For the $\Delta t = 4\,\mathrm{h}$ experiments, the errors are larger than in the $\Delta t = 1\,\mathrm{h}$ case, but with the same hierarchy between the algorithms. In Appendix C, we show that these conclusions are robust when analysing the error on other interface variables.

**Figure 6.** Same as Fig. 5, but with the relative error computed between the final iterated solution and the solution obtained after 2 iterations. See legend for Fig. 4 for the explanation of the x-axis. The ordinate axis is cut at 15 % to make non negligible errors visible.

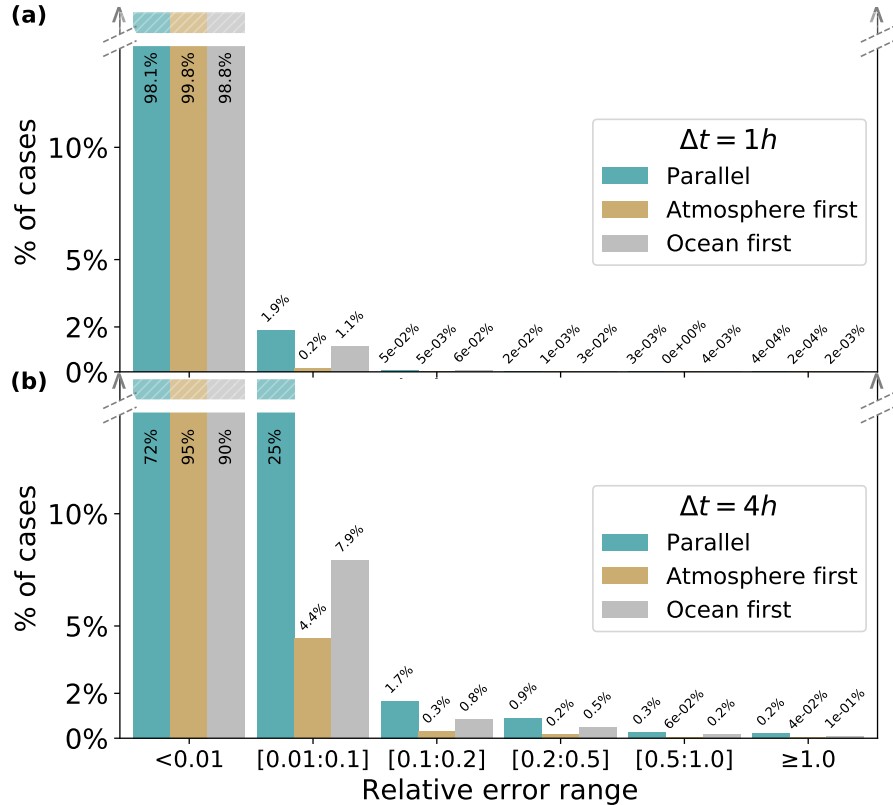

Figure 6 shows the relative error that would remain if we had stopped the Schwarz method at 2 iterations. The histogram shows that for the *parallel* $\Delta t = 1$ h experiment, which is the slowest converging one, non negligible errors ($> 0.01$) account for only about 2 % of the cases. For small coupling periods, a two-iteration Schwarz method strongly improves the solution for the *parallel* algorithm, with only a handful of cases that need more than 5 iterations to reach a small error (less than 0.1, not shown). All these points are at the ice edge, where the convergence is slower. For *parallel* $\Delta t = 4$ h, 25 % of the cases have an error in the range $[0.01, 0.1]$. The number of cases with error larger than 0.1 after 2 iterations amounts to about 3.5 %. This is still a large improvement compared to the non-iterated *parallel* algorithm.

These results are coherent with the theoretical results on Schwarz methods mentioned previously. The *sequential* Schwarz method requires approximately twice less iterations to converge than the parallel algorithm. In IPSL-CM6-SW-VLR, both *sequential* algorithms converge faster than the *parallel* algorithm.

This result is not symmetric: the *sequential atmosphere-first* algorithm converges faster than the *sequential ocean-first* algorithm. We propose two hypotheses to explain this phenomenon. First, the characteristic time scales are longer in the ocean than in the atmosphere, and the diurnal cycle is more marked in the atmosphere than in the ocean. Therefore, using the information

from the ocean on the previous time window to force the atmospheric model on the next time window is probably generally less problematic than doing the opposite. The atmospheric solution after the first half-iteration will then already be quite close to its converged value, and will provide a relevant and synchronized forcing to compute the oceanic solution in the second half-iteration. Second, the better performance of the *atmosphere-first* case can also be linked to the phasing of the solar radiation, which is the only external forcing and constrains the diurnal cycle. In the *parallel* and *ocean-first* cases, the ocean is forced by fluxes, including solar radiation, calculated at the previous coupling period. In the case of *atmosphere-first*, the solar forcing is correctly phased.

## 4.3 The diurnal cycle of the error

Figure 7 plots the SST trend error in function of the roman local time and error classes, for the *parallel* experiments (see figure caption for the definition of the roman local time). The error histograms show a well-defined diurnal cycle with the lowest errors during the night. In both experiments, but mostly for $\Delta t = 4\,\mathrm{h}$, errors are larger at noon than at midnight. The error is maximum after sunset and before sunrise, when the change of the insolation forcing evolves at the fastest pace. This pattern is clear for $\Delta t = 1\,\mathrm{h}$. With $\Delta t = 4\,\mathrm{h}$, the diurnal cycle of insolation is badly resolved, but the diurnal cycle of the error is still present. After sunrise and before sunset, 45 % of cases in time $\times$ space show an error larger than 1 for the $\Delta t = 1\,\mathrm{h}$ case. At these times of the day more than 70 % of the cases show error larger than 0.5, and almost all cases have non negligible errors ($\geq 0.01$). All figures are slightly bigger for the $\Delta t = 4\,\mathrm{h}$ case.

An error larger than 1.0 means that the correction due do the Schwarz method is larger than the solution jump due to the lagged algorithm. In both experiments, the error of the *parallel* algorithm after sunrise and before sunset can affect the most important part of the solution computed by an Earth System model.

## 5 Conclusions and future approaches

Present time algorithms used to couple ocean and atmosphere in state-of-the-art Earth System Model are mathematically inconsistent in all implementation we are aware of. The components are not correctly synchronized with their boundary conditions (Lemarié, 2008; Lemarié et al., 2014). A mathematically consistent Schwarz iterative method has been implemented in the IPSL coupled model to solve the ocean-atmosphere interface. This implementation yields a multiplication of the computing cost by the number of iterations. Although such a method is thus not affordable as is for climate studies, the Schwarz iterative method is used as a reference to evaluate the error made with the *parallel* and the *sequential atmosphere-first* currently used by many ocean-atmosphere modelers. The *sequential ocean-first* has also been tested.

We use the solution obtained with the Schwarz iterative method as a reference to diagnose the error in six experiments, with the three coupling algorithms and two coupling period lengths $\Delta t = 1\,\mathrm{h}$ and $\Delta t = 4\,\mathrm{h}$. In the *parallel* algorithm, the error is quite large, with highest values after dawn and before dusk, when the change of insolation at the top of the atmosphere, the only external forcing, has the highest rate. With the shortest coupling period of $\Delta t = 1h$, 45% of the cases in time $\times$ space

**Figure 7.** Histograms of errors as a function of the roman local time and error classes, for the *parallel* experiments. Left panels (a, b) for $\Delta t = 1\,\text{h}$, right panels (c, d) for $\Delta t = 4\,\text{h}$. Top (a, c) panels show the percentage of cases in space × time in each range of error, as a function of the roman local time. The roman local time is the local time piece-wise stretched : the time from midnight to sun rise is divided in 6 regular pseudo hours. The same method is applied to the intervals sun set to noon, noon to sun set and sun set to midnight. This is similar to the division of the day in the ancient Rome (Wikipedia, 2020). The percentages are computed with respect to the total number of cases for each local time. Bottom panels (b, d) show the number of cases with errors larger than 0.1 (light purple), 0.5 (medium purple with hatches) and 1.0 (dark purple).

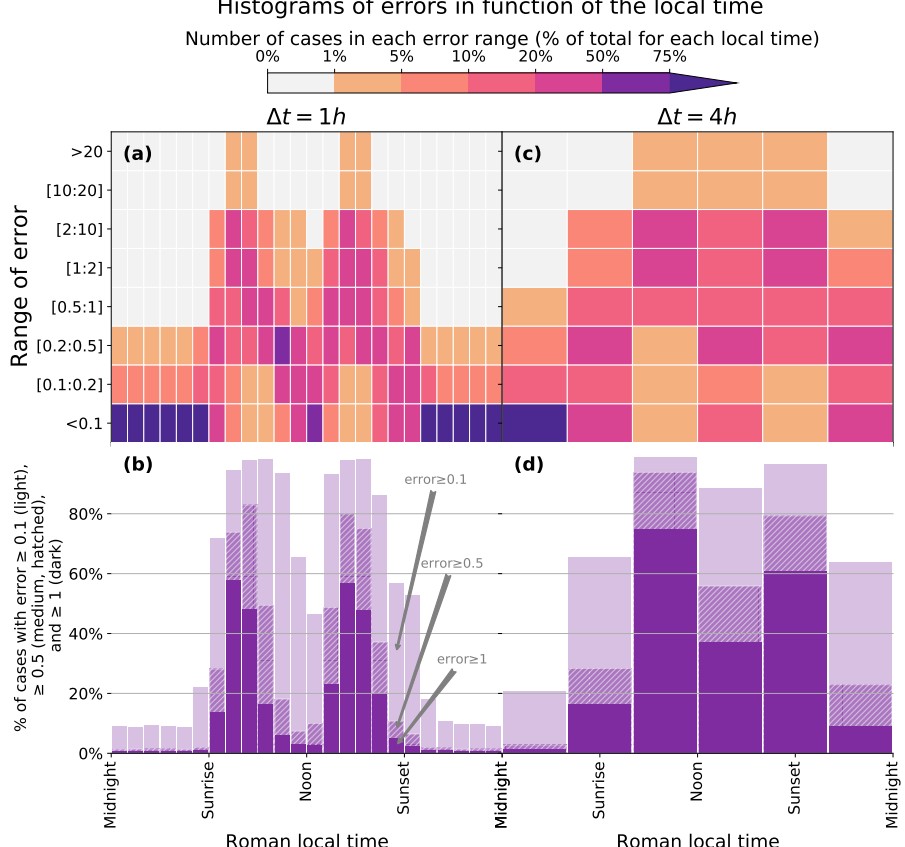

show an error larger than 100% for this periods of the day. That means that for this time of the daily cycle, the solution without Schwarz suffers from a large error in most of the cases. With a larger coupling period, the errors are even larger.

Our analysis shows that implementing *sequential* algorithms are simple ways to strongly reduce the error, with the *atmosphere-first* algorithm showing the best performance. We propose two hypotheses to explain the *atmosphere-first* algorithm performance. First, the atmosphere has shorter characteristic time scales than the ocean, with a more marked diurnal cycle. The atmospheric lower boundary condition evolves slowly, and the atmospheric solution after the first half-iteration is then already quite close to its converged value, and provides a relevant and synchronized forcing to compute the oceanic solution

in the second half-iteration. Second, the better performance of the *atmosphere-first* case can also be linked to the phasing of the solar radiation, which is the only external forcing and constrains the diurnal cycle. In the *parallel ocean-first* case, the ocean is forced by fluxes, including solar radiation, calculated by the atmosphere at the previous coupling period. In the case of *atmosphere-first*, the solar forcing is correctly phased. The *sequential* algorithms, however, have a major drawback. The models do not run concurrently as, while one model is running, the other model waits for its coupling information coming from the one running. This eliminates a level of parallelism, and increases the time to solution of the coupled model, unless a two-coupling-period lag is introduced for the feedback of the ocean on the atmosphere, which increases the time inconsistency of the algorithm.

The error of all algorithms, and particularly of the *parallel* one, can be strongly reduced by performing only two iterations. This is still a huge increase of the computing cost, which is clearly unacceptable. The vast majority of iterative methods have a speed of convergence that is very sensitive to the choice of the initial state. The target is to reduce the number of iterations down to 1 which would mean keeping a classical, non-iterated lagged method. But the idea would be to reduce the error thanks to a judiciously chosen initial state. A first approach could be an extrapolation of the previous time steps. A second approach could be to perform Schwarz iterations on a sub part of the model, to get an improved first guess before running the full model once. It will be effective if we can identify parts of the models that represent only a small part of the calculation cost, but account for a large part of the change of the model state during a coupling period. The coupled vertical turbulent diffusion term of both models, including the computation of turbulent fluxes at the interface, is a possible candidate.

With two iterations, a conservation issue appears with the *parallel* algorithm. The second, and last, iteration of the ocean model uses the fluxes computed by the atmosphere during the first iteration. The atmosphere will get its energy and water balance from the fluxes computed at the second iteration. Both components do not use the same fluxes, which yields a con-servation inconsistency at the interface. This happens when the iterative process is stopped before the full convergence. In this case, the ocean model would have to run one more iteration than the atmosphere to close the energy and water cycle between the model components, at an expense of the computing time.

It is likely that our results observed at the ocean-atmosphere interface can be generalised to other couplings in Earth system models when lagged algorithms are used, like ocean-sea ice, atmosphere-sea ice or atmosphere-soil. These interfaces with rapid variability, especially with dry soil or thin sea ice, can be very sensitive to the coupling algorithm. We did not assess the effect of the errors at the coupling interface on the simulated climate, in terms of means and variability at monthly to multi-decennial time scales. The internal feedbacks in a climate model make the impact uncertain. If the model with the legacy *parallel* coupling scheme computes, for instance, a too high interface temperature at a given coupling period, the atmosphere to ocean heat fluxes of the following coupling period will be reduced accordingly and may partly compensate the error, with a time lag. A modification of the diurnal cycle in both amplitude and phase can be expected. But the error might be somewhat canceled when considering diurnal means, or longer time scales. How will the long term means and variability, which are the properties analyzed by climatologists, be affected? To assess this impact, two ensembles of climate experiments, with and without Schwarz, should be compared. The model with the Schwarz iterative method is currently too expensive for us to

carry out this set of experiments. We will try to reduce this cost before carrying out a comprehensive assessment, mainly by improving the first guess, and limiting the Schwarz method to a few iterations.

To reduce the error, one could simply reduce the coupling period. In IPSLCM6-SW-VLR, the ocean time step is 1 h. Reducing the length of the coupling period implies reducing the ocean time step, and increasing the computer time. With higher resolution, the time step of the ocean or the atmosphere are smaller, and it is possible to couple more often. As with any discretisation, the error decreases with the time step. This should be used cautiously however, as most interface fluxes are computed by bulk formulas. Gross et al. (2018) shows that a $\Delta t_{phys,req}$ time scale is needed for a bulk formulation to be valid. The inputs of the bulk formula, like sea surface temperature, should be averaged over this time scale to minimize the uncertainty (Gross et al., 2018; Large, 2006; Foken, 2006). $\Delta t_{phys,req}$ is usually greater than the model dynamical time step $\Delta t_{dyn}$. This means that reducing the time step is not coherent with the basic assumption made to obtain the bulk formulas, and may yield large error in the flux computation.

## Code and data availability

All code and data relative to this study are available at https://zenodo.org/record/4546183 (Marti et al., 2020). This Digital Object (DOI) Identifier points to three files. Marti-GMD-2020-307_Models.tar.zip is a gzipped tar file of $218\,\mathrm{MB}$ with the model code and scripts needed to run the model (Fortran, C++ and bash). Marti-GMD-2020-307_Figures.zip is a zip file of $3.3\,\mathrm{MB}$ containing the scripts needed to produce the figures: one PyFerret script and seven Jupiter Python Notebooks. Marti-GMD-2020-307_Data.tar.zip is a a gzipped tar file of $18.5\,\mathrm{GB}$ with the model outputs needed to produce the figures.

We give in the following more references for the code used. LMDZ, XIOS, NEMO and ORCHIDEE are released under the terms of the CeCILL license. OASIS-MCT is released under the terms of the Lesser GNU General Public License (LGPL). We used model version IPSLCM6.1.9-LR, which is build from the following model components and utilities (svn branches and tags) :

- NEMO : `branches/2015/nemo_v3_6_STABLE/NEMOGCM`, Tag : 9455

- ORCA1 config : `trunk/ORCA1_LIM3_PISCES`, Tag : 278

- IPSLCM6 : `CONFIG/UNIFORM/v6/IPSLCM6`, Tag : 4313

- ORCHIDEE : `tags/ORCHIDEE_2_0/ORCHIDEE`, Tag : 5661

- OASIS : `branches/OASIS3-MCT_2.0_branch/oasis3-mct`, Tag : 1818

- IOIPSL : `IOIPSL/tags/v2_2_4/src`, Tag : HEAD

- LMDZ : `LMDZ6/branches/IPSLCM6.0.15`, Tag : 3427

- libIGCM : `trunk/libIGCM`, Tag : 1478

- XIOS : `XIOS/branchs/xios-2.5`, Tag : 1550

Model documentation is available at https://forge.ipsl.jussieu.fr/igcmg_doc/wiki/Doc. The code modifications made in IPSL-CM6.1.9-LR to build IPSL-CM6-SW-VLR and implement the Schwarz iterative method are fully documented at https://forge.ipsl.jussieu.fr/cocoa.

## Appendix A: Evaluation of IPSL-CM6-SW-VLR

IPSL-CM6-SW-VLR simulated climate has substantial differences with IPSL-CM5A2-LR, due to the different soil and sea ice models. We present here a short evaluation of the simulated climate of a steady-state pre-industrial simulation. The initial state for the ocean of the atmosphere is taken from the reference IPSL-CM5A2-LR simulation of Sepulchre et al. (2020). For the ice model, LIM2 and LIM3 states are not compatible. In the present case, the sea ice initial state is set to a fixed height of ice where the ocean temperature of the first level (at $5\,\mathrm{m}$ depth) is at the freezing point. The height of ice is $3\,\mathrm{m}$ in the northern hemisphere and $1\,\mathrm{m}$ in the south. On land the albedo parameters of the bucket model was taken from the albedo computed by ORCHIDEE in the reference PREIND simulation of Sepulchre et al. (2020), which follows the CMIP6 intercomparison project of the *piControl* experiment (Eyring et al., 2016). In a first attempt, the model evolves towards a cold state, due to an imbalance of about $-2.8\,\mathrm{W m^{-2}}$ of the radiation budget at the top of the atmosphere (TOA).

The procedure described by Sepulchre et al. (2020) is then used to balance the model heat budget. A parameter controlling the conversion of cloud water to rainfall is tuned to reach a near zero net flux at top of the atmosphere (TOA), with a target of $13.5\,^\circ\mathrm{C}$ for global mean near surface temperature (temperature at $2\,\mathrm{m}$ height). The final TOA heat budget is $0.33\,\mathrm{W m^{-2}}$, with a global mean near surface temperature of $13.3\,^\circ\mathrm{C}$. Figure  A1 shows the simulated sea surface temperature (SST) compared to Sepulchre et al. (2020) and to the World Ocean Atlas (WOA, Locarnini et al., 2013).

As expected from the drastic simplification of the soil model, the performances, in term of simulated climate, of IPSLCM6-SW-VLR are poorer than those of the state-of-the models participating for example in the CMIP6 intercomparison exercise. But as the objective of this study focuses on the evaluation of the Schwarz method, a model with a perfect simulated climate is not necessary. We estimate that a good part of the degradation of this version compared to IPSL-CM5A2 is linked to the soil model.

## Appendix B: Algorithms

The Schwarz loop is intimately embedded in the time step loops of the model. Algorithm 1 shows the normal time loop in a model (ocean or atmosphere). Algorithm 2 shows the model time loop modified to incorporate the Schwarz iteration loop. There is no single way to implement the algorithm. In particular, to implement the possibility to have Schwarz window spanning several time steps, the Schwarz loop should be the outside loop. But this implies more complex changes in the original codes, and we chose the fastest way.

**Figure A1.** Sea surface temperature (SST) difference between the different versions of IPSLCM, and with the World Ocean Atlas. Left: January, right: July. Top : IPSLCM6-SW-LR minus IPSLCM5A2-LR, middle: IPSLCM6-SW-LR minus the World Ocean Atlas (WOA, Locarnini et al., 2013), bottom: IPSLCM5A2-LR minus the World Ocean Atlas. IPSLCM6-SW-LR are averaged over 10 years. IPSLCM5A2-LR are averaged over 100 years.

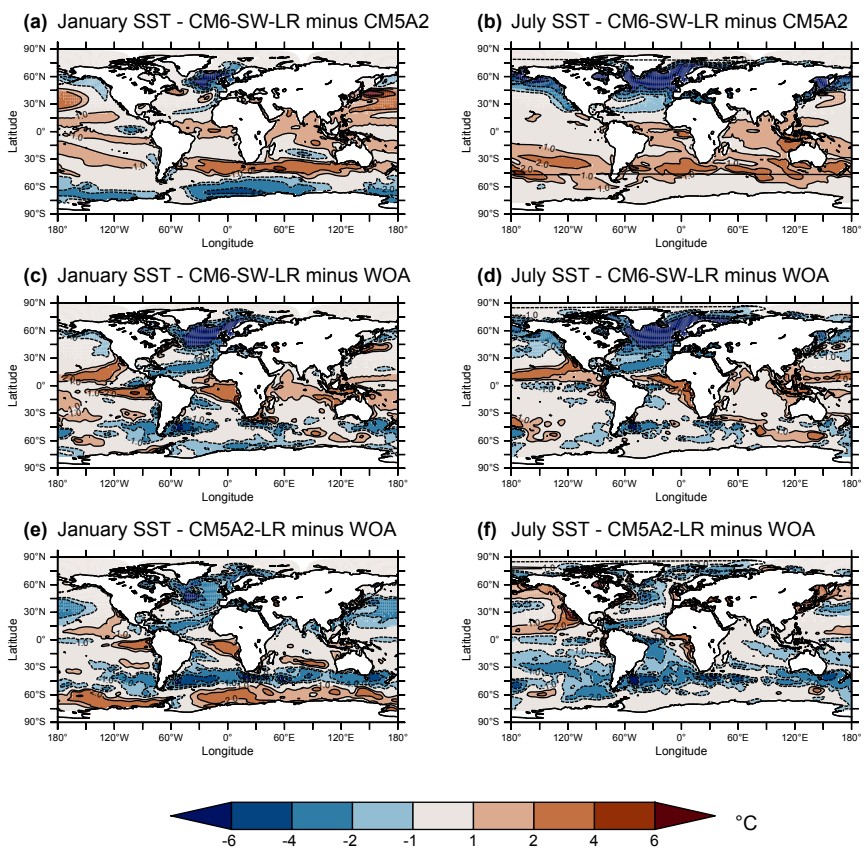

## Appendix C: Are conclusion drawn from sea surface temperature robusts ?

In the main text, we use the sea surface temperature (SST) to diagnose the convergence speed and the error. By construction, the convergence speed is in theory identical for all variables. After SST convergence, the atmosphere uses the same values of SST at each iteration, and by construction compute the same fluxes. Symmetrically, when the fluxes computed by atmosphere have converged, the ocean can do nothing but producing the same SST at each iteration. In practice, the full convergence is not obtained, and a small oscillation remains for all interface variables. As the convergence criterion is somewhat arbitrary, the computation of the number of iterations before convergences can give different value for the different variables.

---

**Algorithm 1** Normal model time stepping

---

$n_{end} \leftarrow$ number of model time steps for the total run

$n_{cpl} \leftarrow$ number of model time steps in one coupling period

```
Initialize model
```

**for** $i_t \leftarrow 1, n_{end}$ **do**

    **if** $i_t \mod n_{cpl} = 1$ **then**

```
        Receive coupling fields ← Other model
```

    **end if**

```
    Run model time step
```

    **if** $i_t \mod n_{cpl} = 0$ **then**

```
        Send coupling fields → Other model
```

    **end if**

**end for**{Loop on $i_t$}

```
Finalize model
```

---

**Algorithm 2** Model time stepping with the Schwarz method

---

$n_{end} \leftarrow$ number of model time steps for the total run

$n_{cpl} \leftarrow$ number of model time steps in one coupling period (and Schwarz window)

$n_{sloops} \leftarrow n_{end}/n_{cpl}$ number of Schwarz windows (i.e. coupling periods) for the total run

```
Initialize model
```

**for** $i_{sloops} \leftarrow 1, n_{sloops}$ **do**

```
    Store model state at first Schwarz iteration before first time step → Memory
```

    // Schwarz loops iteration for the coupling period

    **for** $k_{swr} \leftarrow 1, m_{swr}$ **do**

        **if** $k_{swr} > 1$ **then**

```
            Reset model state to first time step for new Schwarz iteration ← Memory
```

        **end if**

```
        Receive coupling fields ← Other model
```

        **for** $i_t \leftarrow 1, n_{cpl}$ **do**

```
            Run one model time step
```

        **end for**{Loop on $i_t$}

```
        Send coupling fields → Other model
```

    **end for**{Loop on $i_{swr}$}

**end for**{Loop on $i_{swloop}$}

```
Finalize model
```

---

**Figure C1.** Same as Fig. 5, but for non solar heat flux. Relative error of the change of non solar heat flux during a coupling period. The error is computed as the ratio between i) the correction due to the iterative procedure (the jump from green dots to converged solution in grey in Fig. 3) and ii) the solution change between $t$ to $t + \Delta t$ with no Schwarz iteration (the jump from yellow to green dots in Fig. 3). See legend for Fig. 4 for the explanation of the x-axis.

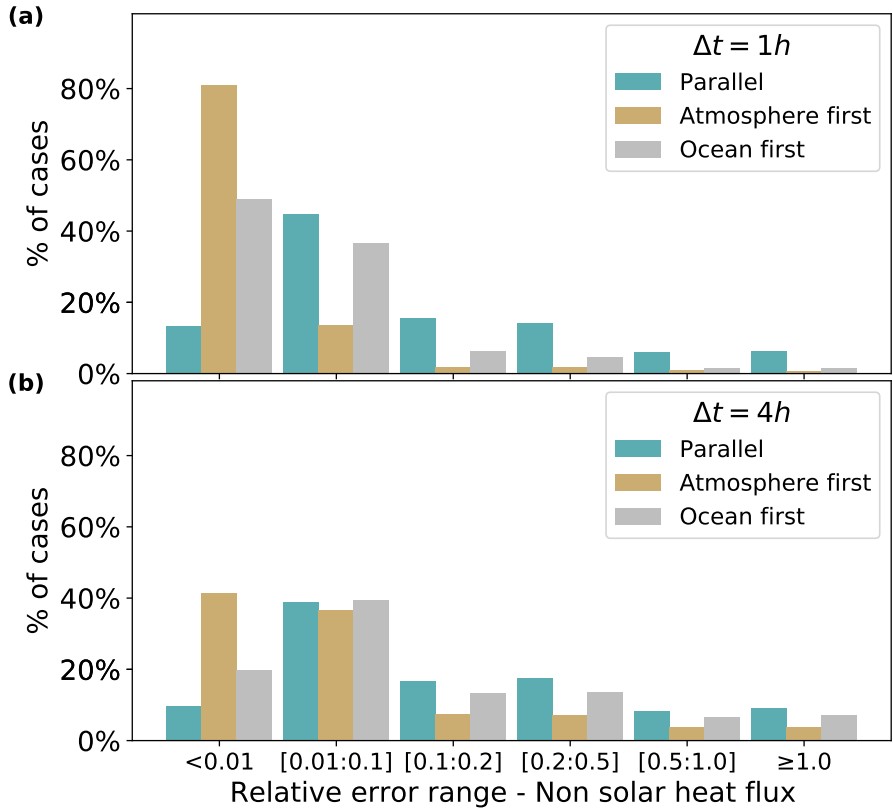

Figure C1 plots the histogram error for the non solar heat flux. This is the same computation than for Fig. 5. The histograms show some differences when compared to SST histograms. However, the main conclusions of the analysis are the same, with large errors for the *parallel* case, and lesser error for the *sequential* cases.

*Author contributions.* Olivier Marti co-designed the study, runs some experiments, made the analysis and wrote the paper with large inputs by Florian Lemarié, Sophie Valcke and Eric Blayo. Sébastien Nguyen helped to design the study, made all the coding to implement the
Schwarz method, run some experiments and made some analysis. Pascale Braconnot co-designed the study. Sophie Valcke brought her expertise in coupling. Florian Lemarié and Eric Blayo designed the mathematical framework and brought their expertise in all mathematical aspects.

*Competing interests.* Authors declare no competing interests.

*Acknowledgements.* This study is part of the ANR project COCOA (https://anr.fr/Projet-ANR-16-CE01-0007). This work was granted access to the HPC resources of TGCC under an allocation made by GENCI (Grand Équipement National de Calcul Intensif, grant 2019-A0040100239). It benefits from the development of the common modelling IPSL infrastructure coordinated by the IPSL climate modeling center (https://cmc.ipsl.fr). Data files were prepared with NCO (netCDF Operators, Zender, 2008, and http://nco.sourceforge.net). Sketches are drawn with LibreOffice (https://www.libreoffice.org). Plots and histograms are produced with Matplolib (Hunter, 2007, and https://matplotlib.org) in Jupyter Python notebooks. Maps where drawn with pyFerret, a product of NOAA's Pacific Marine Environmental Laboratory (http://ferret.pmel.noaa.gov/Ferret). Patrick Brockman and Jean-Yves Peterschmitt brought invaluable help in the realisation of the figures. We thanks the reviewers for their kind yet helpful reviews: the original paper studied the IPSLCM6 legacy *parallel*, and the *sequential* ones where added from their suggestions.

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
