# Peer review of "A Schwarz iterative method to evaluate ocean-atmosphere coupling schemes; implementation and diagnostics in IPSL-CM6-SW-VLR"

_Geoscientific Model Development, 2020_

## Referee Comment (RC1) · Anonymous Referee #1 · 28 Dec 2020

Most existing coupled models suffer from the temporal inconsistencies at the interface between component models. This paper proposes a Schwarz iterative method that can reduce such inconsistencies as well as the corresponding errors. The experimental results demonstrate that the proposed iterative method can converge fast in an ocean-atmosphere coupled model and reveal that the temporal inconsistencies in existing coupled models can produce significant errors. This paper is well written and well structured, and the key idea is clearly presented. In my opinion, the idea and results in this paper deserve wide attentions from the community.

The following are my specific comments and suggestions.

[Figure]

1. It seems that an important pre-condition of the proposed Schwarz iterative method is that the converged solution of coupling fields at model time T1 based on the initial states at model time T0 is the exact coupling fields at T1. It will be welcome to state such a pre-condition and briefly introduce the corresponding theoretical supports.

2. The proposed Schwarz iterative method uses SST for judging convergence. One possible guess is that the convergence speed may be relative to the fields used. For example, SST generally changes slowly in time integration, which may contribute to the fast convergence. So, it will be welcome to evaluate the convergence speed using another field such as wind speed that generally changes fast, and it may be interesting to compare differences of solutions of coupling fields under different convergence variables.

3. The model used in this study is a climate model. It will be welcome to discuss possible application the proposed iterative method in real climate simulations. One possible challenge here is how to make the iterative method not break conservation. Weather forecasting that does not highly depend on conservation may be a potential application (this study uses 5-day simulation actually). It will be welcome to show the differences resulting from the iterative method after a 5-day simulation. Considering the resolution of weather forecasting becomes very fine, it will be welcome to evaluate or discuss the proposed method under finer resolutions.

4. It will be welcome to discuss applications of the proposed method in a model coupling with different frequencies (for example, an atmosphere model uses 1-hour frequency while an ocean model uses 4-hour frequency based on the averaged atmospheric values in each 4 hours), and discuss applications in a complex coupled model with more component models.

5. Many existing coupled models use concurrent coupling between atmosphere and land surface. I believe that this coupling can also benefit from the proposed method. It will be welcome to make a discussion, as land surface states generally change much

faster than ocean states especially at sunrise and sunset.

6. A brief introduction to IPSL-CM6-SW-VLR should be included in the abstract, as it has been included in the title.

7. It will be welcome to provide a table for how to evolve IPSL-CM6-LR to IPSL-CM6-SW-VLR.

8. It will be welcome to provide a figure for the software architecture of the IPSL-CM6-SW-VLR with the iterative method.

9. Line 181: "each coypling time step"=>"each coupling time step".
* * *

---

## Referee Comment (RC2) · Anonymous Referee #2 · 12 Jan 2021

The paper presented is a valuable contribution to the development of coupled AOGCM configurations, presenting a tool for probing the validity of simplifying assumptions that we collectively make in the coupled climate modelling community. The Schwarz iterative method is used to achieve converged ocean-atmosphere state vector and fluxes at the interface. I have have little formal complains, as the paper is in, with one exception, in excellent shape with regards to presentation and clarity.

- My main feedback point is that, while the authors concede that the Schwarz iterative method is impractical for production runs and is envisioned to serve as a validation tool, it is nevertheless compared to production runs with double sided

lag. For validation purposes other groups may currently use a simple one sided lag configuration. In this case the production run eq. (3):

$$\left|\frac{dA}{dt}\right|_t^{t+\Delta t} = F_A(A, \langle O_\Omega \rangle_{t-\Delta t}^t), \quad \left|\frac{dO}{dt}\right|_t^{t+\Delta t} = F_O(O, \langle f_\Omega \rangle_{t-\Delta t}^t) \qquad (1)$$

would be compared to:

$$\left|\frac{dA}{dt}\right|_t^{t+\Delta t} = F_A(A, \langle O_\Omega \rangle_t^{t+\Delta t}), \quad \left|\frac{dO}{dt}\right|_t^{t+\Delta t} = F_O(O, \langle f_\Omega \rangle_{t-\Delta t}^t) \qquad (2)$$

In order to solve this system the models are forced into alternate execution, typically with the flux computing atmosphere going second. In the very first timestep the ocean model assumes zero surface fluxes and updates the ocean surface state based on internal dynamics only, while the atmosphere waits. Once the ocean is done the atmosphere updates and the ocean waits. The models computational performance also degrades by about a factor two compared to double lagged production runs, however the implementation is very simple. For OASIS coupled models we simply need to set lag=0 in the configuration file. For other models such as the ECMWF IFS the same is achieved by having the ocean model as a subroutine in the atmospheric code and using the same cores.

I think this paper would benefit from comparing your Schwarz iterative method with the single sided lag. Regardless of the outcome, whether you find that single sided lag already gets you close to the convergence of Schwarz step #2, or not, your work is highly valuable. Either you can provide the community with a better tool for the validation of coupled models, or you can validate the existing validation method and undergird its use. Just which one it is, is not clear to me after reading through.

- A second and smaller feedback point is that section 3.2 is somewhat detached from the rest of the paper and I wonder if it could not be an appendix. I'm also

wondering why 5 day long simulations are compared climatology. Was it an ensemble of 5 day simulations?

- line 182: spelling of coypling should be coupling

- Figure 4: values and dots don't line up. (E.g. 2 on the axis is actually 2,5 rounded down)

I recommend the paper for major review. While method presented appears solid, more targeted validation could improve the relevance further.

---

## Author Comment (AC1) · 19 Mar 2021

**Response to reviewer comments on "A Schwarz iterative method to evaluate ocean- atmosphere coupling schemes. Implementation and diagnostics in IPSL-CM6-SW-VLR" by Olivier Marti et al.**

*Geosci. Model Dev. Discuss.* https://gmd.copernicus.org/preprints/gmd-2020-307 *- 2020*

**Anonymous Referee #1**

Most existing coupled models suffer from the temporal inconsistencies at the interface between component models. This paper proposes a Schwarz iterative method that can reduce such inconsistencies as well as the corresponding errors. The experimental results demonstrate that the proposed iterative method can converge fast in an ocean- atmosphere coupled model and reveal that the temporal inconsistencies in existing coupled models can produce significant errors. This paper is well written and well structured, and the key idea is clearly presented. In my opinion, the idea and results in this paper deserve wide attentions from the community.

We thank the Reviewer for the appreciation and the thoughtful comments. In the following we answer each specific point (in blue).

Note that in the corrected manuscript, formulas are now in black and white. Colours in the text are not accepted by Copernicus publication. We apologize for the slight loss of readability.

As suggested by reviewer #2, we ran and analysed experiments with a one sided lag method. The manuscript has substantial changes to incorporate these.

The following are my specific comments and suggestions.

1. It seems that an important pre-condition of the proposed Schwarz iterative method is that the converged solution of coupling fields at model time T1 based on the initial states at model time T0 is the exact coupling fields at T1. It will be welcome to state such a pre-condition and briefly introduce the corresponding theoretical supports.

In simple linear models, the unicity of the converged solution is proven. It does not depends on the initial guess which can be chose arbitrarily. A good choice of the initial improves the convergence speed. In our case, the models are strongly non linear. There are possibly several possible solutions, an the converged solution may depend on the initial guess. A relevant choice of the initial guess is then important. Using the converged solution of the previous Schwarz window is our best 'educated guess'. We add this explanation at the end of part 2.3.

2. The proposed Schwarz iterative method uses SST for judging convergence. One possible guess is that the convergence speed may be relative to the fields used. For example, SST generally changes slowly in time integration, which may contribute to the fast convergence. So, it will be welcome to evaluate the convergence speed using another field such as wind speed that generally changes fast, and it may be interesting to compare differences of solutions of coupling fields under different convergence variables.

We add a comment about convergence speed

By construction, the convergence speed is in theory identical for all variables. After SST convergence, the atmosphere uses the same values of SST at each iteration, and computes the same fluxes. Symmetrically, when the fluxes computed by atmosphere have

converged, the ocean can do nothing but producing the same SST at each iteration. In practice, the full convergence is not obtained, with a small oscillation of the values. As the convergence criterion is somewhat arbitrary, the computation of the number of iterations before convergences can give different value for the different variables

We add appendix C to show the error histogram for the non solar heat flux. We didn't use the windspeed as suggested. Wind speed at 10 m height is a diagnostic variables computed by assuming a logarithmic profile between the first layer (at 100 m height) and the ground, depending of the stability. In the current model version, it is transmitted to the ocean, but use only to compute $CO_2$ fluxes, and not for the dynamical part. We prefer to use a prognostic variable. Appendix C show that the main conclusions of the paper are not sensitive to the choice of the interface variable analysed.

3. The model used in this study is a climate model. It will be welcome to discuss possible application the proposed iterative method in real climate simulations. One possible challenge here is how to make the iterative method not break conservation. Weather forecasting that does not highly depend on conservation may be a potential application (this study uses 5-day simulation actually). It will be welcome to show the differences resulting from the iterative method after a 5-day simulation.

We realized the comparison method is badly explained in the text: we do not compare experiences with and without Schwarz. The experiences without Schwarz described in the table are not used. The table has been simplified. We only use the simulations with Schwarz. And we study the surface temperature trend term over one coupling time step. At each iteration of Schwarz, the model compute this trend term. At the first iteration, we have the trend term that the model calculates with the legacy method (lagged coupling). We can then compare it with the trend term obtained with Schwarz (after convergence). This comparison of the two terms is done on the same trajectory of the model. Since we are working on a single model trajectory, it is not necessary to make ensembles of simulations to know whether the difference in climate or weather obtained is significant.

We have improve this explanation in the text.

Considering the resolution of weather forecasting becomes very fine, it will be welcome to evaluate or discuss the proposed method under finer resolutions.

It seems hard to have a response. If the resolution of the models is increased, the time steps decrease and it is possible to couple more often. As with any discretisation, the error decreases with the time step. On the other hand, if the increase in resolution increases the rapid variability (hourly to sub-hourly), the effect is to increase the error. As shown by Gross et al. (Gross et al., 2018), an internally required time scale $\Delta t_{phys,req}$ needs to be assumed for the parameterization scheme (bulk formulation) to be valid. This means that reducing the time step is not coherent with the basic assumption made to obtain the bulk formulas. We add some comment at the end of the conclusion.

4. It will be welcome to discuss applications of the proposed method in a model coupling with different frequencies (for example, an atmosphere model uses 1-hour frequency while an ocean model uses 4-hour frequency based on the averaged atmospheric values in each 4 hours), and discuss applications in a complex coupled model with more component models.

This might be done with a Schwarz window which encompass the longest coupling period. We add a few words on this at the end of section 2.4.

5. Many existing coupled models use concurrent coupling between atmosphere and land surface. I believe that this coupling can also benefit from the proposed method. It will be welcome to make a

discussion, as land surface states generally change much faster than ocean states especially at sunrise and sunset.

We put a few words in the conclusion.

6. A brief introduction to IPSL-CM6-SW-VLR should be included in the abstract, as it has been included in the title.

Done.

7. It will be welcome to provide a table for how to evolve IPSL-CM6-LR to IPSL-CM6- SW-VLR.

Done

8. It will be welcome to provide a figure for the software architecture of the IPSL-CM6- SW-VLR with the iterative method.

9. Line 181: "each coypling time step"=>"each coupling time step".

Done

**Anonymous Referee #2**

The paper presented is a valuable contribution to the development of coupled AOGCM configurations, presenting a tool for probing the validity of simplifying assumptions that we collectively make in the coupled climate modelling community. The Schwarz iterative method is used to achieve converged ocean-atmosphere state vector and fluxes at the interface. I have have little formal complains, as the paper is in, with one exception, in excellent shape with regards to presentation and clarity.

We thank the Reviewer for the appreciation and the thoughtful comments. In the following we answer each specific point (in blue).

In the corrected manuscript, formulas are now in black and white. Colours in the text are not accepted by Copernicus publication. We apologize for the slight loss of readability.

• My main feedback point is that, while the authors concede that the Schwarz iterative method is impractical for production runs and is envisioned to serve as a validation tool, it is nevertheless compared to production runs with double sided lag. For validation purposes other groups may currently use a simple one sided lag configuration. In this case the production run eq. (3):

$$\left.\left|\frac{d\mathcal{A}}{dt}\right.\right|_t^{t+\Delta t} = \mathbf{F}_{\mathcal{A}}(\mathcal{A}, \langle\mathcal{O}\rangle_{t-\Delta t}^t), \qquad \left.\left|\frac{d\mathcal{O}}{dt}\right.\right|_t^{t+\Delta t} = \mathbf{F}_{\mathcal{O}}(\mathcal{O}, \langle\mathbf{f}\rangle_{t-\Delta t}^t) \tag{1}$$

would be compared to:

$$\left.\left|\frac{d\mathcal{A}}{dt}\right.\right|_t^{t+\Delta t} = \mathbf{F}_{\mathcal{A}}(\mathcal{A}, \langle\mathcal{O}\rangle_t^{t+\Delta t}), \qquad \left.\left|\frac{d\mathcal{O}}{dt}\right.\right|_t^{t+\Delta t} = \mathbf{F}_{\mathcal{O}}(\mathcal{O}, \langle\mathbf{f}\rangle_{t-\Delta t}^t) \tag{2}$$

In order to solve this system the models are forced into alternate execution, typically with the flux computing atmosphere going second. In the very first time step the ocean model assumes zero surface fluxes and updates the ocean surface state based on internal dynamics only, while the atmosphere waits. Once the ocean is done the atmosphere updates and the ocean waits. The models computational performance also degrades by about a factor two compared to double lagged production runs, however the implementation is very simple. For OASIS coupled models we simply need to set lag=0 in the configuration file. For other models such as the ECMWF IFS the

same is achieved by having the ocean model as a subroutine in the atmospheric code and using the same cores.

I think this paper would benefit from comparing your Schwarz iterative method with the single sided lag. Regardless of the outcome, whether you find that single sided lag already gets you close to the convergence of Schwarz step #2, or not, your work is highly valuable. Either you can provide the community with a better tool for the validation of coupled models, or you can validate the existing validation method and undergird its use. Just which one it is, is not clear to me after reading through.

It is indeed relatively easy to implement a single sided method in our version of the model. We were very interested by your idea, and we run experiments with one sided lagged methods, with atmosphere first and ocean first. The results are very different with the different methods. We redesigned the paper to include these results. Thank you very much for this suggestion that enlarge the scope of the paper.

• A second and smaller feedback point is that section 3.2 is somewhat detached from the rest of the paper and I wonder if it could not be an appendix.

Yes, that seems very relevant for the ease of reading, and we have moved this part to appendix A.

I'm also wondering why 5 day long simulations are compared climatology. Was it an ensemble of 5 day simulations?

The validation of IPSLCM6-SW-VLR is made on 10 years means compared to climatology. This is has been added in the legend of the figure A1.

• line 182: spelling of coypling should be coupling

Corrected

• Figure 4: values and dots don't line up. (E.g. 2 on the axis is actually 2,5 rounded down)

Corrected.

I recommend the paper for major review. While method presented appears solid, more targeted validation could improve the relevance further.